# The Developmental Trajectory of Motor Competence of Children That Lived the COVID-19 Confinement Period: A Four-Year Follow-Up Study in Portuguese Children

**DOI:** 10.3390/jfmk7030064

**Published:** 2022-08-31

**Authors:** Aida Carballo-Fazanes, Luis Paulo Rodrigues, Rui Silva, Vitor P. Lopes, Cristian Abelairas-Gómez

**Affiliations:** 1CLINURSID Research Group, Psychiatry, Radiology, Public Health, Nursing and Medicine Department, Universidade de Santiago de Compostela, 15782 Santiago de Compostela, Spain; 2Simulation, Life Support, and Intensive Care Research Unit (SICRUS) of the Health Research Institute of Santiago de Compostela (IDIS), 15706 Santiago de Compostela, Spain; 3Instituto Politécnico de Viana do Castelo, Escola Superior Desporto e Lazer de Melgaço, 4900-347 Viana do Castelo, Portugal; 4Research Center in Sports Performance, Recreation, Innovation and Technology (SPRINT), 4960-320 Melgaço, Portugal; 5Research Center in Sports Sciences Health and Human Development (CIDESD), 5000-801 Vila Real, Portugal; 6Instituto Politécnico de Bragança, Campus de Santa Apolónia, 5300-223 Bragança, Portugal; 7Faculty of Education Sciences, Universidade de Santiago de Compostela, 15782 Santiago de Compostela, Spain

**Keywords:** COVID-19 lockdown, physical activity, motor competence, motor competence assessment

## Abstract

Children’s motor competence (MC) was negatively affected by the COVID-19 pandemic; however, possible chronic effects have not been studied. Therefore, the aim of this study was to examine the possible impact of the forced lack of physical activity (PA) during the COVID-19 lockdown on children’s MC two years later. The motor competence of sixty-seven healthy children (7.4–12.2 years old) was assessed using the Motor Competence Assessment (MCA). All participants completed the MCA tests at two different moments (before and after the COVID-19 lockdown), four years apart. The mean values after the COVID-19 lockdown for all participants on the subscales and on the Total MCA are lower, but no significant changes were found when controlling for gender and age (*p* > 0.05 in all analyses). However, a significant decrease was found in the Locomotor subscale in boys (*p* = 0.003). After dividing the participants into three age groups, the youngest also suffered a decrease in the Locomotor subscale (*p* < 0.001) and their Total MCA (*p* = 0.04). In addition, those participants who had a higher MC at baseline decreased their scores for the Locomotor (*p* < 0.001) and Manipulative (*p* < 0.001) subscales, and for the Total MCA (*p* < 0.001). In conclusion, the younger children and the more motor proficient did not fully recover from the negative effects of the pandemic lockdown after two years.

## 1. Introduction

The development of motor competence (MC) over the growing years is seen as a cornerstone for a more active and healthier lifestyle across all ages, in accordance with the development and performance of human movement. MC can be defined as a person’s ability to be proficient in a broad range of locomotor, stability, and manipulative gross motor skills [1]. MC is a key factor for the development of motor skills, fostering the mechanisms of learning, adaptation, and transfer, and it is expected to help with the proficiency of novel motor tasks throughout the lifespan [2]. Furthermore, MC has been associated with multiple developmental outcomes, including physical health [3,4,5] and psychological, social-emotional, and cognition/achievement [6,7,8,9].

Several constraints have been recognized in the literature as influential in the development of MC across different developmental ages, namely the biological (e.g., gender, height, BMI, maturation, physical fitness, etc.); sociocultural (e.g., ethnicity, culture, education, family, etc.), and physical activity (PA) (e.g., movement opportunities, sports, physical education, sedentary time, etc.) [9]. Relative to the last, studies usually assess PA and sedentary levels of the individuals to compare or relate to MC levels and so to infer the effect on MC [10,11,12]. The ecological problem with this approach is that we cannot really parcel out the real effects of PA or inactivity. Participation in PA should be seen as a multifactorial question, and children that engage less in PA or more in sedentary behavior are surely not similar to the more physically active ones. The only way to find the real effect of physical (in)activity is to randomly compare similar individuals constrained to different movement opportunities or to the absence of PA. Although conditions like these are usually unethical to use with children, the recent COVID-19 lockdown has provided the opportunity to study the effect of forced physical inactivity on the development of children. 

The COVID-19 pandemic, first notified in December 2019 in China and officially recognized as a pandemic on 11 March 2020, had worldwide repercussions and impact due to its aggressiveness and fast spread. One of the strategies recommended for slowing down its dissemination was to lockdown entire families in their houses, closing schools and public parks, and canceling all sports and physical activities. Thus, people living in house lockdown for several weeks to months were less mobile and, consequently, the spread of the virus was reduced. In the lockdown period, physical inactivity and sedentary lifestyles increased, especially among children and adolescents [13,14,15,16]. As a consequence, and unsurprisingly, the development of health-related fitness [14] and MC in schoolchildren [17] and preschool children [18] were negatively affected. 

Previous studies have shown that in periods out of school, children are more likely to engage in unhealthy behaviors such as sedentary behavior [19,20] and that this negatively affects their MC [21]. Nevertheless, few studies have studied the acute effects of the COVID-19 lockdown on MC, and no studies have yet analyzed the possible chronic effects after two years or more. Therefore, questions arise as to whether the negative effects of inactivity are permanent and whether children’s resilience will enable them to return to the expected developmental trajectories of CM. Studying the chronic effects of the COVID-19 lockdown on MC trajectories may allow professionals to outline strategies to avoid possible regressions on MC during unexpected inactivity. 

The aim of this study was to examine the impact of the lack of PA during the COVID-19 lockdown on children’s developmental trajectories of MC two years later. This is an opportunistic study in the sense that it was not foreseen or previously organized. Because of that, and the restrictions that took place during and after the COVID-19 lockdown period and the ones that remained enforced during the two years after the first lockdown event, no information was collected during or immediately after it. Even so, there is no doubt about the overall restrictions that all children suffered because of the pandemic event, which represented a one and only period of forced inactivity for all children. They had to remain indoors, constrained to their families’ houses, with no direct contact with their peers. All community sports activities were forbidden, and all outdoor spaces for physical activities were closed. Furthermore, in these initial times of the pandemic, the fear of contagion led families and children to self-restrict their contacts and time outside, even when possible. Hence, and despite the lack of more information that could be valuable to fully understand this abnormal developmental period for each child, we believe it interesting to report the characteristics of the developmental trajectories of MC found in this period. Thus, the present study predictions were threefold: (1) the expected normative development of MC was impaired two years after the COVID-19 lockdown period; (2) the developmental trajectories of MC, two years after the COVID-19 lockdown, were different according to the age and gender of the children; and (3) the developmental trajectories of MC, two years after the COVID-19 lockdown, were different according to the baseline MC values of the children.

## 2. Materials and Methods

### 2.1. Participants

This observational study involved a convenience sample of 67 apparently healthy children (39.7% girls), from 7.4 to 12.2 decimal years at baseline. They were recruited from a public primary school in Melgaço (Viana do Castelo, Portugal). Data were collected at two different moments, four years apart: Moment (M) 1—April 2018 (pre-pandemic situation) and M2—April 2022 (post-pandemic situation). All these children experienced a 3-month lockdown from 18 March to 3 May 2020, however, restrictions on physical activity remained in effect until the end of June. 

The inclusion criteria for participants were not to be diagnosed with a motor development disorder and to be between 7 and 12 years of age at M1. Exclusion criteria were suffering from any illness that would make it impossible to perform the tests. Participants were divided into three age groups for the analyses. Group 1: age 7–8.5 years, Group 2: age 9–10.5 years, and Group 3: age 10.5–12 years (at the initial time, M1). They were also divided into three groups according to the degree of proficiency in motor competence. Those children ranked in the lowest tercile of the 2018 MCA percentiles were considered Low MC and those ranked in the highest tercile were considered High MC. The rest were considered Average MC. Written informed consent was obtained for the legal parents/Guardians to sign, as well as the verbal consent of each child, prior to the tests. The study protocol was approved by the scientific committee of the School of Sports and Leisure (Code: CTC-ESDL-CE003-2017). 

### 2.2. Instruments

Motor competence was assessed using the Motor Competence Assessment (MCA) [22]. MCA is a valid and reliable product-oriented motor test consisting of two tests from each subscale: stability, locomotor, and manipulative [2]. Normative values according to gender and age are available from 3 to 23 years of age [22]. Table 1 lists the specific characteristics of each test. The scores of the MCA tests were converted into percentile values according to the MCA normative values adjusted to age and gender. Subscale scores were calculated by the average of the two percentile values of the two tests. The total MCA score was calculated by the average of all percentile scores of the six tests.

Anthropometric measurements (weight and height) were measured with a scale and measuring road (*Seca, Hamburg, Germany*) twice, non-consecutively, and the mean of both was recorded. Then, Body Mass Index (BMI) was obtained by applying the following equation: weight (kg) divided by height^2^ (m) [kg/m^2^].

### 2.3. Procedures

Children’s motor competence was assessed at two points in time, four years apart: M1 (April–May 2018) and M2 (March 2022). These assessments are part of a longitudinal study that takes part with the Melgaço School cluster. In this study, each child should be assessed every two years, however, because of COVID-19, the 2020 assessment period was not possible, and we were only allowed by the school to proceed with the regular assessments in 2022. The tests of the MCA were conducted in a sports hall during the school’s regular schedule, and at least one experienced researcher supervised each test (see Table 1 for more information). The participants, in small groups (approximately 5 children) were given a verbal explanation and a practical demonstration of the test to be performed. They then made a practical trial followed by the test trials. The best score from the trial attempts was considered for the analyses. Motivational feedback was provided to all children in the same way, and no specific information about correct or incorrect performance was provided during the test trials.

### 2.4. Statistical Analysis

According to the aims of the study, results were analyzed according to gender, age, and M1 (initial value or baseline) of the participants. Means and standard deviation were initially used to describe the results by condition. Repeated measures ANOVA were used to assess the developmental trajectories from M1 to M2 for each Subscale and Total MCA according to gender, age, and M1. Age and gender were included in the analysis as factors to control for possible bias. To interpret the magnitude of the effect size, the following criteria were adopted [23]: <0.20 trivial, 0.20–0.50, small; 0.50–0.80, medium; and >0.80, large. All analyses were performed using the SPSS statistical package version 25.0 (SPSS Inc., Chicago, IL, USA). A significance level of *p* < 0.05 was considered.

## 3. Results

Children had at M1 a mean age of 9.3 ± 1.5 years, height 135.9 ± 10.1 cm, weight 35.1 ± 10.6 kg, and Body Mass Index (BMI) 18.7 ± 3.8 kg/m^2^ and at M2 a mean age of 13.3 ± 1.5 years, height 158.3 ± 10.7 cm, weight 53.4 ± 14.8 kg, and BMI 21.1 ± 4.5 kg/m^2^. 

The results are presented in Table 2 to 4 according to Gender, Age groups, and baseline (M1) values for each subscale.

Although the average values of M2 for all participants on the Locomotor and Manipulative Subscales and Total MCA are lower than in M1, no significant changes were found between the two moments, when controlling for gender and age (*p* > 0.05 in all analyses). But when divided by gender (columns Boys and Girls on the Table) we found a significant decrease (7.7 percentiles on average) in the Locomotor Subscale values for boys (*p* = 0.003), although the effect size is small (0.22). All other MCA Subscales and Total scores remained similar between M1 and M2 (*p* > 0.05) (see all participants in Table 2).

When the developmental trajectories of MC were partitioned by age group participants (Table 3), we found that the older Group 3 did not present any significant change from M1 to M2 after controlling for the possible effect of gender. 

The middle age Group 2 showed a significant increment in Stability (*p* = 0.009) relative to an average change of 10.5 percentile points but indicated no changes in all other MCA measures. In the younger Group 1, we found a significant negative change of 9.2 percentile points in the Locomotor Subscale (*p* < 0.001); but more importantly, we found that this age group had a negative change in its Total MCA after controlling for gender. In all cases, the effect size was small.

After testing for the effects of different baseline levels of MC (Table 4), it was found that children with Low MC before the COVID-19 lockdown tended to improve their percentile scores on all Subscales and Total MCA, and particularly, they significantly improved their Stability (*p* = 0.046) and Manipulative scores (*p* = 0.015). On the other end of the MC spectrum, children that presented a higher MC at baseline generally decreased all their MCA scores, and they did it in a statistically significant way for the Locomotor (*p* < 0.001) and Manipulative subscales (*p* < 0.001), and for the Total MCA (*p* < 0.001).

## 4. Discussion

The main purpose of this study was to understand if the physical inactivity period forced by the COVID-19 lockdown would impact the trajectories of the development of MC in children. The MCA was used to test for MC two years before and after the lockdown period. The results are presented as normative values of the MCA Subscales and Total (percentile scores adjusted to age and gender), meaning that the expected outcome in a typical situation would be for children to reasonably maintain their MC scores in the same developmental percentile channel. In consequence, after the forced COVID-19 inactivity, a significant decrease in percentile values was expected.

Relative to the results of all samples, no significant change was found (Table 2) when the results were controlled for the possible cohort effect of the age groups and gender (*p* > 0.05 in all analyses). Although the Locomotor subscale average percentile values changed from 32.1 to 25.1 (between M1 and M2) these changes were not statistically significant. The results seem counterintuitive and do not concord with previous studies [17,18] that found significant differences between moments (before, during, or immediately after the imposed movement restriction period). However, it seems evident, from the limited available literature, that MC suffers negative effects from the forced motor inactivity during the lockdown [17,18] due to movement behaviors acquired in this period of school closures, activity limitation, and physical distance such as long exposure times to screens and less physical activity [24,25]. Still, no information exists on the follow-up of such negative effects. Our results seem to suggest that children’s resilience in this matter can be responsible for some of the developmental rebound [26].

When analyzing the gender differences, it was found that only boys were shown to have their developmental trajectories of the Locomotor subscale negatively affected two years after COVID-19. All other MCA values remained unchanged, showing that both genders were equally able to rebound from the negative effects of COVID-19 after a two-year period.

Regarding the next step of our analysis, we tried to see if children at different moments of their individual development would sense different effects of forced inactivity. That is, when at different stages of their motor skill development, children would suffer different effects from the lack of movement. In general (see Table 3), all three age groups tested showed a decrease in the average score of the subscales and total MCA. However, the younger group that lived the pandemic situation with a mean age 9.9 years (range from 9.9 to 10.3 years) showed the biggest negative effects with a negative deviation from the typical developmental trajectories of the Locomotor (*p* < 0.001) and the Total MCA (*p* = 0.04) scores. A negative change was also found in the Stability subscale for Group 2 (mean age of 11.6 years), increasing the M2 score, and no significant deviations from the predicted normalized developmental pathway were found for the older age group (13.8 years). These findings seem to suggest that children at different developmental moments of their MC had different responses to the forced absence of movement, with younger children (in less advanced phases of motor skill learning and development) not being able to overcome the problems created by the lack of movement. This is in line with the idea that motor opportunities during some specific critical periods are important to progress into more high levels of MC, and that the lack of motor stimulation during these critical periods can delay or impair normal motor development [27,28].

Our last hypothesis was that the levels of MC would influence the developmental trajectories of MC after the inactivity period. The analysis was made to compare groups of different levels of MC at the baseline. The ones with Low MC represent the children classified in the lower tercile of the MCA percentile scores in 2018, while the High MC group represents the children that scored in the highest tercile of the percentile scores. The rationale for this prediction was that different intensities and volumes of motor stimulation are necessary to achieve an optimal stimulus depending on the MC level already acquired. 

Our results seem to support this prediction because it was in the High MC group (after controlling for age group and gender cohort effects) that the effect of the COVID-19 lockdown showed the most negative effects. The High MC group was the only one where all mean values were lower at M2, and children showed statistically significant negative changes in two of the subscales (Locomotor and Manipulative) and at the MCA total (*p* > 0.05 in all analyses). On the other end of the MC proficiency, the lower MC group was able to maintain similar values for the MCA total, and it presented positive developmental changes in the Stability and Locomotor subscales. These differential results associated with the proficiency levels of MC are probably explained by the fact that higher levels of motor stimulation are needed to maintain higher levels of MC, while for children with low MC proficiency, less motor stimulation or the episodical lack of it, do not promote the immediate change of their developmental trajectories. The fact of having two subscales where the Low MC group showed a positive change in their MC trajectories is more difficult to explain, but maybe it was linked to some extra (not habitual) motor opportunities that children were exposed to after the final of the confinement period. In fact, after the lockdown period, families were probably more attentive to the problem of sedentary activities, and maybe that led to an increment in these children’s physical activities and motor opportunities that, given their low MC levels, could constitute an appropriate stimulus for MC improvement. 

Some of the differences in the trajectories of MC could be influenced by the somatic characteristics of the sample and their distribution according to the MCA percentile groups, but, in general, no statistically significant differences in BMI or height were found between percentile groups of MCA, for each age group separated by gender (data not shown). Only the Low MC boys in age group 3 in 2018 had higher BMI than their peers, but this difference was not found in 2022. These results support the idea that changes in MC occurred independently of the somatic features of the children.

Our results do not fully support the first postulated prediction that the expected normative development of MC remained impaired two years after the COVID-19 lockdown period. The second prediction, which suggested possible differences in the developmental trajectories of MC, according to age and gender, was partially endorsed by the data, since in general, boys and girls showed similar behavior, but age was a marker of the developmental period and was shown to influence the effects of COVID-19 forced inactivity, with younger children showing more difficulty in rebounding from the effects of COVID-19 forced inactivity, probably given their position in fewer advance phases of MC development. 

The study also supports our third prediction that the MC level before COVID-19 can differentiate the effect of the lack of physical activity and motor opportunities that children experienced during the COVID-19 lockdown. In this case, children located in the highest percentiles of MC were probably more affected by the decrease in regular PA due to the pandemic situation, and their scores worsened to a greater extent than those with initially lower percentiles, most of whom either maintained or even improved their scores. Probably because of that, they had more difficulty rebounding from the lack of stimulation, and two years after, they still experienced a delay in their MC development trajectories. 

A major limitation of this work is the lack of MC evaluation just before and after the COVID-19 event. As such, we assume that the status of each child just before the lockdown was similar to the MC assessment taken two years before in April 2018 and that all children suffered an effective detrimental effect on their MC development as a consequence of the forced inactivity of the lockdown. These assumptions are not without risk but are supported by the studies that showed the decrement in MC immediately after the COVID-19 lockdown event [17,18]. Another limitation was the absence of maturational status assessment and physical activity (before and during) that could provide a better interpretation of the results, given the age span characteristics of the sample. Because of the unexpected events that took place with the rise of the pandemic situation, no information on the real characterization of each child’s life during the COVID-19 first lockdown and successive events was possible, and that also constitutes a major limitation. Some of the children may even be directly affected by the illness with all the possible (and even now not fully understood) principal and secondary effects. Since the COVID-19 pandemic was impossible to foresee, we tried to work with the best possible information we had included in the data collection to take advantage of this unique event where children were forcefully restricted to a sedentary state for a long period of time. As previously explained in the methods section, our access to further individual data relative to this period was constrained because of the pandemic-specific laws in place, and the school’s (and parent’s) authorization to proceed with the longitudinal study. Even so, and because the expected impact of the force lockdown has been described as massive [15,16,17,18], a major and recognizable effect on the developmental trajectories of the children MC has to be acknowledged, and that was the main goal of this study with all its limitations. The follow-up of these children, and the collection of more information on the individual conditions of how each family and children lived during the lockdown, along with a retrospective morphological assessment, can bring more important information to complete the whole picture of the effects of forced sedentary moments in children.

The findings of this study should be taken into account for periods in which, for some reason, PA cannot be practiced on a regular basis, and the promotion of alternatives is recommended and necessary because of the negative effect of physical inactivity on MC, especially for younger children and those children with greater motor proficiency.

## 5. Conclusions

The present study revealed that after two years from the forced inactivity period of 3 months in 2020, children between 9.4 and 14.2 years of age generally showed they were able to rebound back to their predicted developmental trajectories of motor competence. Their age and proficiency at the time of the sedentary event can be deleterious to the recovery from the loss of motor opportunities and stimulation. The younger and the more motor proficient children did not fully recover from the negative effects of the pandemic lockdown after two years. 

## Figures and Tables

**Table 1 jfmk-07-00064-t001:** Tests included in the Motor Competence Assessment.

		*Material* Procedure
**Stability tests**	Jumping sideways	*Rectangular surface (100 cm length × 60 cm width) divided by a small wooden beam (60 cm length × cm high × 2 cm width)* Jump laterally with both feet together (simultaneously) as fast as possible for 15 s. Each correct jump (two feet together, without touching outside the rectangle and without stepping on the wooden beam) scores 1 point. The best result over 2 trials is considered.
	Shifting Platforms	*Two wooden platforms (25 cm × 25 cm × 2 cm with four 3.7 cm feet at the corners)* Move sideways alternately on two wooden platforms for 20 s, making as many transitions as possible. Each successful transfer is scored with 2 points (1 point for moving the platform sideways; 1 point for moving the body to the platform). The best result over 2 trials is considered.
**Locomotor tests**	Standing Long Jump	*Measuring tape* Jump as far as possible by landing with both feet simultaneously without falling back. The distance (in cm) measured between the starting line and the place where the heel lands closest to the starting line was recorded. The best result over 3 trials is considered.
	10 m Shuttle Run	*Two lines (100 cm × 5 cm) 10 m apart and two rounded blocks.* Run from the starting line to the second line, pick up a block and place it on the starting line, and go back to the second line to pick the second block. The test ends when the participant crosses the start/finish line carrying the second block, and the time in seconds is recorded. The best result over 2 trials is considered.
**Manipulative tests**	Throwing ball velocity	*Tennis ball (diameter: 6.5 cm; weight: 57 g) (children between 3–10 years old). Baseball ball (diameter: 7.3 cm; weight: 142 g) (Children 11-years-old and older). Velocity radar gun (e.g., Pro II Stalker radar gun).* Throw a ball towards the wall with maximum speed with the dominant hand from a line 6 m away. The speed (m/s) is recorded with the velocity radar gun. The best result over 3 trials is considered.
	Kicking ball velocity	*Soccer ball no. 3 (circumference: 62 cm, weight: 350 g) (children between 3–8 years old). Soccer ball no. 4 (circumference: 64 cm, weight: 360 g) (Children 9-years-old and older). Velocity radar gun (e.g., Pro II Stalker radar gun).* Kick a ball at maximum speed against the wall with the dominant foot from a line 6 m away. The speed (m/s) is recorded with the velocity radar gun. The best result over 3 trials is considered.

**Table 2 jfmk-07-00064-t002:** MCA percentile values for Subscales and Total scores, by gender, and comparison between M1 (2018) and M2 (2022) data.

	All Participants (n = 67)				Boys (n = 41)				Girls (n = 26)			
	M1 (2018)	M2 (2022)			M1 (2018)	M2 (2022)			M1 (2018)	M2 (2022)		
	*M (SD)*	*M (SD)*	*p*	*d*	*M(SD)*	*M(SD)*	*p*	*d*	*M (SD)*	*M (SD)*	*p*	*d*
Stability	50.7 (21.5)	54.6 (24.6)	0.700	0.002	47.0 (22.4)	46.7 (23.1)	0.963	0.000	56.5 (18.9)	66.9 (22.0)	0.714	0.006
Locomotor	32.1 (13.4)	25.1 (12.7)	0.054	0.062	29.4 (15.0)	21.7 (12.5)	0.003	0.218	36.2 (9.5)	30.4 (11.5)	0.480	0.022
Manipulative	41.5 (21.2)	40.5 (22.2)	0.150	0.034	41.5 (22.1)	42.4 (26.6)	0.689	0.004	41.4 (20.1)	37.4 (21.6)	0.097	0.115
MCA total	41.5 (13.9)	40.1 (14.6)	0.187	0.030	39.3 (15.1)	36.9 (14.4)	0.257	0.036	44.8 (11.4)	44.9 (17.8)	0.376	0.034

MCA: Motor Competence Assessment; Boys and Girls-ANOVA Repeated Measures with Age Group as a factor (NS for all ANOVAS); Total-ANOVA Repeated Measures with Age Group and Gender as a factor (NS for all ANOVAS).

**Table 3 jfmk-07-00064-t003:** MCA percentile values for Subscales and Total scores, by Age Group, and comparison between M1-2018 and M2-2022 data.

	Age Group 1 (n = 30)				Age Group 2 (n = 24)				Age Group 3 (n = 13)			
	M1 (2018)	M2 (2022)			M1 (2018)	M2 (2022)			M1 (2018)	M2 (2022)		
	*M (SD)*	*M (SD)*	*p*	*d*	*M (SD)*	*M (SD)*	*p*	*d*	*M (SD)*	*M (SD)*	*p*	*d*
Stability	50.1 (18.9)	50.9 (22.7)	0.750	0.004	49.7 (22.3)	60.2 (28.5)	0.009	0.269	54.1 (26.6)	52.6 (20.5)	0.394	0.067
Locomotor	32.6 (13.4)	23.4 (9.6)	<0.001	0.387	31.3 (12.3)	26.0 (15.3)	0.128	0.102	32.7 (16.6)	28.1 (14.6)	0.806	0.007
Manipulative	37.6 (22.8)	34.1 (16.8)	0.290	0.040	40.7 (18.4)	42.3 (26.7)	0.699	0.007	51.6 (20.1)	51.9 (21.0)	0.151	0.178
MCA total	40.09 (13.9)	36.2 (11.4)	0.04	0.143	41.1 (12.3)	43.8 (16.8)	0.313	0.049	46.2 (17.1)	43.3 (16.1)	0.294	0.121

MCA: Motor Competence Assessment; Boys and Girls-ANOVA Repeated Measures with Age Group as a factor (NS for all ANOVAS); Total-ANOVA Repeated Measures with Age Group and Gender as a factor (NS for all ANOVAS).

**Table 4 jfmk-07-00064-t004:** MCA percentile values for Subscales and Total scores, by MC level at the baseline (M1), and comparison between M1-2018 and M2-2022 data.

	Low MC				Average MC				High MC			
	M1 (2018)	M2 (2022)			M1 (2018)	M2 (2022)			M1 (2018)	M2 (2022)		
	*M (SD)*	*M (SD)*	*p*	*d*	*M(SD)*	*M(SD)*	*p*	*d*	*M (SD)*	*M (SD)*	*p*	*d*
Stability	27.5 (9.5)	38.4 (25.7)	0.046	0.184	47.8 (8.8)	53.2 (23.1)	0.164	0.086	74.7 (10.0)	70.7 (14.3)	0.196	0.075
Locomotor	18.3 (6.9)	21.1 (11.7)	0.158	0.093	31.2 (5.6)	22.4 (10.0)	0.001	0.394	47.6 (5.6)	32.3 (13.7)	<0.001	0.566
Manipulative	17.9 (7.9)	25.6 (15.2)	0.015	0.262	39.9 (5.5)	42.5 (21.1)	0.536	0.018	65.5 (11.7)	52.5 (21.6)	<0.001	0.412
MCA total	27.1 (6.5)	31.5 (11.7)	0.50	0.179	39.8 (4.9)	40.4 (15.4)	0.829	0.002	57.7 (6.4)	48.5 (11.5)	<0.001	0.520

MCA: Motor Competence Assessment; ANOVA Repeated Measures with Age Group and Gender as factors (only effect of Age Group for High Locomotor *p* = 0.043; ES = 0.309).

## Data Availability

Contact the correspondent author for data information.

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
