# Peer review of "The Developmental Trajectory of Motor Competence of Children That Lived the COVID-19 Confinement Period: A Four-Year Follow-Up Study in Portuguese Children"

_jfmk, 2022, doi:10.3390/jfmk7030064_

Round 1

Reviewer 1 Report (Previous Reviewer 2)

No others comments

Reviewer 2 Report (Previous Reviewer 1)

Thank you for this new and modified manuscript version. All the limits have been presented and discussed. Thank you

This manuscript is a resubmission of an earlier submission. The following is a list of the peer review reports and author responses from that submission.

Round 1

Reviewer 1 Report

The manuscript is about the examination of the impact of lack of physical activity during the lockdown on children’s motor competence. Although the topic is interesting and could be of interest for the community, its present important limitations in the methodology.

Major comments

M2 was performed 2 years after the lockdown, this is an important limitation of this study. Two years is a long period. From line 236 to line 240 this aspect is considered, but it is not monitored with an appropriate evaluation in this study. In limitation it is highlighted, but according to my opinion it is not enough, data are necessary to understand this aspect. Motor competence depends on the level of physical activity practice and 2 years is a too long period because the effects of the lockdown can be monitored. My suggestion to make this study interesting is to try to collect information from the parents of the children asking them information on how they dealt with the period after the lockdown and how much physical activity / sport they proposed to their children in comparison to the period before the lockdown.

Minor comments

Line 49-63: please, provide references for this paragraph.

Line 77: please, avoid to use questions within the text.

Line 91: please, specify how it was assessed the healthy status of the children.

Line 107: please provide, if there is, the validity, reliability and feasibility of the Motor Competence Assessment.

Please, avoid to use sex, it is better gender

Author Response

Reviewer 1:

The manuscript is about the examination of the impact of lack of physical activity during the lockdown on children’s motor competence. Although the topic is interesting and could be of interest for the community, its present important limitations in the methodology. 

Major comments

M2 was performed 2 years after the lockdown, this is an important limitation of this study. Two years is a long period. From line 236 to line 240 this aspect is considered, but it is not monitored with an appropriate evaluation in this study. In limitation it is highlighted, but according to my opinion it is not enough, data are necessary to understand this aspect. Motor competence depends on the level of physical activity practice and 2 years is a too long period because the effects of the lockdown can be monitored. My suggestion to make this study interesting is to try to collect information from the parents of the children asking them information on how they dealt with the period after the lockdown and how much physical activity / sport they proposed to their children in comparison to the period before the lockdown.

Answer: Thank you very much for your interesting reflection. Unfortunately, and since the Covid-19 epidemic was not expected, this study does not correspond to a precise design study, and because of that, valuable information is missing. This fact is considered in the limitations of the study. The option could be to try to find retrospective information on each family daily life characteristics during the Covid-19, however, we consider that if physical activity is not recorded objectively and with reliable instruments, obtaining these data through families (and after a so long period of time) could be a significant bias for the study due to the subjectivity of the data because of the time that has passed since the lockdown.

Minor comments

Line 49-63: please, provide references for this paragraph.

Answer: Thank you very much for the suggestion, the following references were added:

Robinson, L.E.; Stodden, D.F.; Barnett, L.M.; Lopes, V.P.; Logan, S.W.; Rodrigues, L.P.; D'Hondt, E. Motor Competence and its Effect on Positive Developmental Trajectories of Health. Sports Med. 2015, 45, 1273-1284. doi: 10.1007/s40279-015-0351-6.

D’Hondt, E.; Deforche, B.; Gentier, I.; De Bourdeaudhuij, I.; Vaeyens, R.; Philippaerts, R.; Lenoir, M.A. A longitudinal analysis of gross motor coordination in overweight and obese children versus normal-weight peers. Int. J. Obes. 2013, 37, 61–67. doi: 10.1038/ijo.2012.55. 

Lopes, V.P.; Stodden, D.F.; Bianchi, M.M.; Maia, J.A.; Rodrigues, L.P. Correlation between BMI and motor coordination in children. J. Sci. Med. Sport 2012, 15, 38–43.

De Meester, A.; Stodden, D.; Brian, A.; True, L.; Cardon, G.; Tallir, I.; Haerens, L. Associations among Elementary School Children’s Actual Motor Competence, Perceived Motor Competence, Physical Activity and BMI: A Cross-Sectional Study. PLoS ONE 2016, 11, e0164600.

Line 77: please, avoid to use questions within the text.

Answer: Thank you for this suggestion. The questions were replaced by the following text:

“Therefore, questions arise as to whether the negative effects of inactivity are permanent and whether children's resilience will enable them to return to the expected developmental trajectories of CM” (Line: 78-80)

Line 91: please, specify how it was assessed the healthy status of the children.

Answer: Families and/or teachers were asked about any disorder of the children related to motor development or any other pathology that would impede the performance of the tests or could alter the results.

Line 107: please provide, if there is, the validity, reliability and feasibility of the Motor Competence Assessment.

Answer: Many thanks for the consideration, the recommended information on line 109 of the reviewed manuscript and the following supporting reference have been added:

Rodrigues LP, Cordovil R, Luz C, Lopes VP. Model invariance of the Motor Competence Assessment (MCA) from early childhood to young adulthood. J Sports Sci. 2021;39(20):2353-2360. doi: 10.1080/02640414.2021.1932290.

Please, avoid to use sex, it is better gender

Answer: Thank you very much for the consideration, the term "sex" was replaced by "gender" throughout the manuscript.

Reviewer 2 Report

In this study the authors examine the impact of the forced lack of physical activity during the COVID-19 lockdown on children's motor competence, two years later.

Although the study has the potentiality of being shared with the scientific community, I believe that the manuscript would benefit from a major revision with the attempt to better support their experimental setting.

1. Abstract: they should start with a first paragraph describing the background. This section should outline the following information:

-        What is already known about the subject, related to the paper in question

-        What is not known about the subject and hence what the study intended to examine (or what the paper seeks to present)

2. The theoretical framework is scarce, they should clearly describe the scientific evidence that supports the hypothesis they have raised.

3. A lot of necessary information is missing in methods section:

-      -  What were inclusion and exclusion criteria?

-       - More information should be provided about the participants’ characteristics.

-    - vAnthropometric measurements and physical tests presuppose a protocol. This element is missing from the methodological description, which may imply an impossibility of replicating the study due to a lack of clarity in this regard.

-       - Experimental procedures should be better defined

-        - The intervention protocol should be better described.

   4. The Discussion should be enriched with the existing theory. The authors should clearly describe the scientific evidence that supports their findings.

   5. I would like to see more of the practical implications. Based on the analyzed variables, how the authors intend to use their findings?

 6. The references are correct but weak and incomplete, thus they should be enriched. Moreover, it would be appropriate to include the DOI to all references.

Kind regards

Author Response

Reviewer 2:

In this study the authors examine the impact of the forced lack of physical activity during the COVID-19 lockdown on children's motor competence, two years later.

Although the study has the potentiality of being shared with the scientific community, I believe that the manuscript would benefit from a major revision with the attempt to better support their experimental setting.

  1. Abstract: they should start with a first paragraph describing the background. This section should outline the following information:

- What is already known about the subject, related to the paper in question

- What is not known about the subject and hence what the study intended to examine (or what the paper seeks to present)

Answer: Thank you very much for this suggestion. This information was added to the abstract with the following text:

“Children's motor competence (MC) was negatively affected by the COVID-19 pandemic. However, possible chronic effects have not been studied. Therefore, the aim of this study (…)” (Line: 22-23)

  1. The theoretical framework is scarce, they should clearly describe the scientific evidence that supports the hypothesis they have raised.

Answer: Thank you for your consideration. As mentioned in this manuscript, there is limited scientific evidence on the topic. To the best of our knowledge, the effects of the covid-19 pandemic on children’s motor competence have only been studied in two studies (mentioned in the introduction of the manuscript, line 75 of the revised document) but neither of them studied the medium to long-term effects as is the case in our study (after two years). However, following your suggestion and because of the possible similarity of the lockdown period with other periods when children are out of school and their habits become more sedentary, the following paragraph and references have been added:

Previous studies have shown that in periods out of school, children are more likely to engage in unhealthy behaviours such as sedentary behaviour and that this negatively affects their MC.” (Line: 76-78)

Carrel, A. L.; Clark, R. R.; Peterson, S.; Eickhoff, J.; Allen, D. B. School-Based Fitness Changes Are Lost during the Summer Vacation. Arch. Pediatr. Adolesc. Med., 2007, 161 (6), 561–564. https://doi.org/10.1001/archpedi.161.6.561.

Hesketh, K. R.; Lakshman, R.; van Sluijs, E. M. F. Barriers and Facilitators to Young Children’s Physical Activity and Sedentary Behaviour: A Systematic Review and Synthesis of Qualitative Literature. Obes. Rev., 2017, 18 (9), 987–1017. https://doi.org/10.1111/obr.12562.

Vandorpe, B.; Vandendriessche, J.; Lefevre, J.; Pion, J.; Vaeyens, R.; Matthys, S.; Philippaerts, R.; Lenoir, M. The KörperkoordinationsTest Für Kinder: Reference Values and Suitability for 6-12-Year-Old Children in Flanders. Scand. J. Med. Sci. Sport., 2011, 21 (3), 378–388. https://doi.org/10.1111/j.1600-0838.2009.01067.x.

  1. A lot ofnecessary information is missing in methods section:

Answer: Thank you for your suggestions to improve this section, please find below the specific comments

- What were inclusion and exclusion criteria? 

Answer: The inclusion and exclusion criteria were included in the Participants’ Methods Section:

The inclusion criteria for participants were not to be diagnosed with a motor development disorder and to be between 7 and 12 years of age at M1. Exclusion criteria were suffering from any illness that would make it impossible to perform the tests.” (lines:104-106).

- More information should be provided about the participants’ characteristics. 

Answer: Anthropometric data of the participants were included at the beginning of the Results Section:

“Children had at M1 a mean age of 9.3±1.5 years, height 135.9±10.1 cm, weight 35.1±10.6 kg, and Body Mass Index (BMI) 18.7±3.8 kg/m2 and at M2 a mean age of 13.3±1.5 years, height 158.3±10.7 cm, weight 53.4±14.8 kg, and BMI 21.1±4.5 kg/m2.”

- Anthropometric measurements and physical tests presuppose a protocol. This element is missing from the methodological description, which may imply an impossibility of replicating the study due to a lack of clarity in this regard. 

 Answer: Detailed information on the measurement of anthropometric data was added in the instrument section.

Anthropometric measurements (weight and height) were measured with a scale and measuring road (Seca, Hamburg, Germany) twice non-consecutively and the mean of both was recorded. Then, Body Mass Index (BMI) was obtained by applying the following equation: weight (kg) divided by height2 (m) [kg/m2].”

Information on the Motor Competence Assessment (MCA) tests is given in table 1 and in the sections on instruments and procedures (methodology).

- Experimental procedures should be better defined

- The intervention protocol should be better described.

Answer: This has been improved in the manuscript. Although this is an observational study at two points in time, two years before and after the closure of COVID-19 where there was no intervention of any kind, the study design was missing from the manuscript and has been added in the methods section (line 97).

  1. The Discussion should be enriched with the existing theory. The authors should clearly describe the scientific evidence that supports their findings.

Answer: Thank you for this suggestion. As mentioned above, the particular and recent situation of the covid 19 pandemic means that few studies have focused on the study of its effects on MC and this is the first to look at chronic effects. However, further references and arguments have been added to the discussion to enrich this section.

Guan, H.; Okely, A. D.; Aguilar-Farias, N.; del Pozo Cruz, B.; Draper, C. E.; El Hamdouchi, A.; Florindo, A. A.; Jáuregui, A.; Katzmarzyk, P. T.; Kontsevaya, A.; et al. Promoting Healthy Movement Behaviours among Children during the COVID-19 Pandemic. Lancet Child Adolesc. Heal., 2020, 4 (6), 416–418. https://doi.org/10.1016/S2352-4642(20)30131-0.

Wang, G.; Zhang, Y.; Zhao, J.; Zhang, J.; Jiang, F. Mitigate the Effects of Home Confinement on Children during the COVID-19 Outbreak. Lancet, 2020, 395 (10228), 945–947. https://doi.org/10.1016/S0140-6736(20)30547-X.

  1. I would like to see more of the practical implications. Based on the analyzed variables, how the authors intend to use their findings?

Answer: According to this suggestion, the following paragraph was added at the end of the discussion section:

The findings of this study should be taken into account for periods in which for some reason PA cannot be practiced on a regular basis, and the promotion of alternatives is recommended and necessary because of the negative effect of physical inactivity on MC, especially for younger children and those children with greater motor proficiency.” (Line 284-287)

  1. The references are correct but weak and incomplete, thus they should be enriched. Moreover, it would be appropriate to include the DOI to all references.

Answer: Thank you for this suggestion. The DOI was added to all references that have it. In addition, the following references were included to enrich the manuscript:

D’Hondt, E.; Deforche, B.; Gentier, I.; De Bourdeaudhuij, I.; Vaeyens, R.; Philippaerts, R.; Lenoir, M.A. A longitudinal analysis of gross motor coordination in overweight and obese children versus normal-weight peers. Int. J. Obes. 2013, 37, 61–67. doi: 10.1038/ijo.2012.55. 

Lopes, V.P.; Stodden, D.F.; Bianchi, M.M.; Maia, J.A.; Rodrigues, L.P. Correlation between BMI and motor coordination in children. J. Sci. Med. Sport 2012, 15, 38–43.

De Meester, A.; Stodden, D.; Brian, A.; True, L.; Cardon, G.; Tallir, I.; Haerens, L. Associations among Elementary School Children’s Actual Motor Competence, Perceived Motor Competence, Physical Activity and BMI: A Cross-Sectional Study. PLoS ONE 2016, 11, e0164600.

Carrel, A. L.; Clark, R. R.; Peterson, S.; Eickhoff, J.; Allen, D. B. School-Based Fitness Changes Are Lost during the Summer Vacation. Arch. Pediatr. Adolesc. Med., 2007, 161 (6), 561–564. https://doi.org/10.1001/archpedi.161.6.561.

Hesketh, K. R.; Lakshman, R.; van Sluijs, E. M. F. Barriers and Facilitators to Young Children’s Physical Activity and Sedentary Behaviour: A Systematic Review and Synthesis of Qualitative Literature. Obes. Rev., 2017, 18 (9), 987–1017. https://doi.org/10.1111/obr.12562.

Vandorpe, B.; Vandendriessche, J.; Lefevre, J.; Pion, J.; Vaeyens, R.; Matthys, S.; Philippaerts, R.; Lenoir, M. The KörperkoordinationsTest Für Kinder: Reference Values and Suitability for 6-12-Year-Old Children in Flanders. Scand. J. Med. Sci. Sport., 2011, 21 (3), 378–388. https://doi.org/10.1111/j.1600-0838.2009.01067.x.

Guan, H.; Okely, A. D.; Aguilar-Farias, N.; del Pozo Cruz, B.; Draper, C. E.; El Hamdouchi, A.; Florindo, A. A.; Jáuregui, A.; Katzmarzyk, P. T.; Kontsevaya, A.; et al. Promoting Healthy Movement Behaviours among Children during the COVID-19 Pandemic. Lancet Child Adolesc. Heal., 2020, 4 (6), 416–418. https://doi.org/10.1016/S2352-4642(20)30131-0.

Wang, G.; Zhang, Y.; Zhao, J.; Zhang, J.; Jiang, F. Mitigate the Effects of Home Confinement on Children during the COVID-19 Outbreak. Lancet, 2020, 395 (10228), 945–947. https://doi.org/10.1016/S0140-6736(20)30547-X.

Round 2

Reviewer 1 Report

Thank you for your consideration in some of my points but the main limitation of the study related to the long period between the two data collections is still present. I am of the opinion that two years for a children are an extremely long period. Furthermore, too many factors can influence the results. I strongly suggest to perform a retrospective study and integrate it with the present results or to deeply investigate the literature to support the findings with more studies.

Reviewer 2 Report

No comments